# Squalamine reverses age-associated changes of firing patterns of myenteric sensory neurons and vagal fibres

Karen-Anne McVey Neufeld[1], Yu-Kang Mao[1], Christine L. West[1,2], Matthew Ahn[1], Hashim Hameed[1], Eiko Iwashita[1], Andrew M. Stanisz[1], Paul Forsythe [3], Denise Barbut[4], Michael Zasloff [4,5✉] & Wolfgang A. Kunze [1,2,6✉]

Vagus nerve signaling is a key component of the gut-brain axis and regulates diverse physiological processes that decline with age. Gut to brain vagus firing patterns are regulated by myenteric intrinsic primary afferent neuron (IPAN) to vagus neurotransmission. It remains unclear how IPANs or the afferent vagus age functionally. Here we identified a distinct ageing code in gut to brain neurotransmission defined by consistent differences in firing rates, burst durations, interburst and intraburst firing intervals of IPANs and the vagus, when comparing young and aged neurons. The aminosterol squalamine changed aged neurons firing patterns to a young phenotype. In contrast to young neurons, sertraline failed to increase firing rates in the aged vagus whereas squalamine was effective. These results may have implications for improved treatments involving pharmacological and electrical stimulation of the vagus for age-related mood and other disorders. For example, oral squalamine might be substituted for or added to sertraline for the aged.

[1] Brain-Body Institute, McMaster University, Hamilton, ON, Canada. [2] Department of Biology, McMaster University, Hamilton, ON, Canada. [3] Department of Medicine, 569 Heritage Medical Research Center, University of Alberta, Edmonton, AB, Canada. [4] Enterin, Inc., Philadelphia, PA, USA. [5] MedStar–Georgetown Transplant Institute, Georgetown University School of Medicine, Washington, DC, USA. [6] Department of Psychiatry and Behavioural Neurosciences, McMaster University, Hamilton, ON, Canada. ✉email: maz5@georgetown.edu; kunzew@mcmaster.ca

ntestinal vagal afferents are the rapid information superhighway through which chemosensory information is relayed from the gut to the brain. Moreover, perhaps because of its huge sensory surface, the gut supplies the majority of vagal afferent fibres to the brain[1]. Despite this, little is known about how vagal afferent function alters in the aged although dystrophic anatomical changes have been observed in vagal afferents innervating the intestine[2].

The primary target of vagal innervation in the murine gut is the myenteric plexus, while direct vagal innervation of the mucosa and submucosa are sparse or absent[1]. Indeed the densest innervation of the intestinal epithelial layer cells[3,4] is supplied by the enteric nervous system, which provides more than 90% of sensory neuropeptide containing fibres to the mucosal layer. While there are several functional classes of neurons within the myenteric plexus, only one class, the intrinsic primary afferents (IPANs) is both chemo- and mechanosensitive serving as an intramural sensory gatekeeper relaying the signals originating from luminal contents to the afferent vagus nerve[5].

It has been proposed, based on age-related changes in calcium-binding proteins and neurotransmitter content, that IPANs are most susceptible to neurodegeneration when aged are compared to young animals[6,7]. Given the crucial gatekeeper role that the IPANs have in relaying information to the afferent vagus, any study of age-related changes in vagal function must be accompanied by a study of age-related functional changes in myenteric IPANs.

Background mesenteric nerve fibre discharge is diminished in aged compared to young adult humans[8] and this finding has been replicated in C57BL/6 mice for which constitutive mesenteric multiunit spiking was significantly reduced in aged (18–24 mo) compared to young (3 mo) animals[9]. Also, we have previously reported that the background vagal afferent firing rate is reduced in aged CD-1 mice by 62% compared to young CD-1 and that intraluminal infusion of the aminosterol squalamine dilactate could return vagal firing rates to those seen young animals[10]. Since electrical vagal stimulation has been used clinically to treat depressive disorders[11] it is possible that a diminished or altered resting vagal afferent firing rate and pattern in the elderly contributes to the significant occurrence of behavioural depression in this group[12].

The class of neuron most likely responsible for the reduction in resting vagus basal firing rate with ageing is the IPAN. With respect to how IPANs signal to the afferent vagus, we have previously reported[5] that, for young mice, neurotransmission is via an IPAN soma to vagal afferent terminal nicotinic synapse that we have termed the intramural sensory synapse.

The aged murine enteric nervous system displays dystrophic neurons[13] and degenerating sensory neuropeptide containing nerve fibres[14]. Aged enteric neurons also accumulate lipofuscin in their somata[14]. However despite this evidence for anatomical and neurochemical changes there have been no physiological recordings made from aged IPANs or other enteric neurons. It is thus hardly possible to know how aged IPANs are functionally altered compared to young IPANs. In view of the increased human median age in the developed world and the importance of the enteric nervous system in gut-to-brain transmission it is critical to investigate an animal model of the aged enteric nervous system. It is also important to identify drugs whose ingestion might reduce the functional effects of ageing.

The constitutive firing rate of vagal fibres innervating the small intestine is lower for aged compared to young mice (see above) but vertebrate neurons encode information by firing patterns and intervals not just firing rates. Despite this, it is not known whether there exists a specific age-associated vagal firing pattern (ageing code). [15]Vagal firing patterns can be described by 4 parameters,

mean interspike interval (MII), burst duration (BD), gap duration (GD) and intraburst interval (IBI), that describe firing intervals with respect to action potential bursts and the overall mean interspike interval[15]. This raises the question whether an ageing code incorporating these parameters can be determined by comparing vagal firing patterns between old and young animals and whether the addition of luminal squalamine to the small intestine of aged mice can reduce or eliminate the ageing code. This would be important because the afferent vagus projects monosynaptically via the nucleus of the solitary tract to the hypothalamic arcuate nucleus[16] that releases growth hormone releasing hormone which programs somatic ageing[17], thus removal of the gut-derived vagus ageing code by squalamine might alter the functional effects of somatic ageing.

In addition, acute treatment of major depressive disorders with selective serotonin reuptake inhibitor (SSRI) antidepressants such as sertraline may reduce therapeutic efficacy in the aged[12,18]. To discover whether peripheral vagal mechanisms might contribute to this effect we applied intraluminal sertraline to the aged intestine to establish whether the SSRI can still facilitate afferent vagal firing as it does in young mice[15]. We also tested if squalamine, which evokes an antidepressant vagal code in young animals[15], is more resistant to ageing in this regard than sertraline.

For the present article, mixed extracellular action potentials (multiunit responses) were recorded from the jejunal mesenteric nerve bundles, and extracellular discharge from individual single vagal fibres (single units) identified by their unique action potential shape and amplitude[15,19].

Patch clamp whole-cell recordings were made from the myenteric plexus using ex vivo segments of mouse jejunum as described in Mao et al.[20], and passive and active physiological properties of IPANs examined.

We identified the defining temporal characteristics of an ageing code for both IPANs and afferent vagal fibres when neurons from aged (18–24 mo) were compared to those from young (3 mo) animals. We also determined that the vagal ageing code was suppressed by application of squalamine.

## Results

**Young and old IPANs differ physiologically**. We have previously reported[10] that colonic migrating motor complexes for aged CD-1 mice have reduced velocity and frequency compared to similar motor complexes recorded from young mice. It is well-known that such colon and small intestine propulsive motor patterns are generated by the enteric nervous system[21, pp. 81-89] and disappear when the enteric nervous system is absent[22], when IPANs are selectively silenced pharmacologically[23] or by point mutation-induced inhibition of protein kinase A in IPANs[24].

In the current study the electrical characteristics of the IPANs of aged mice were compared with those of young ones. We identified all IPANs electrophysiologically as myenteric neurons that possessed a hump on the descending phase of their action potential and had a post-action potential slow afterhyperpolarisation (sAHP)[20,25] For passive cell membrane characteristics, the resting membrane potential ($V_m$,) input resistance ($R_{in}$,) and leak conductance ($g_{leak}$,) differed between young (green) and aged (brown) myenteric IPANs (Fig. 1a). However, neither membrane capacitance ($C_m$) nor action potential (AP) width at half height above baseline ($AP_{1/2width}$) differed statistically between young and aged IPANs. $V_m$ increased by 24% from -55 m$V$ (effect size $\eta^2_p = 0.46$), $R_{in}$ decreased by 35% from 265 M$\Omega$ ($\eta^2_p = 0.31$) and $g_{leak}$ increased by 76% from 4.1 nS ($\eta^2_p = 0.50$) when aged were compared to young neurons (Fig. 1a). For active membrane characteristics young IPANs showed greater excitability than

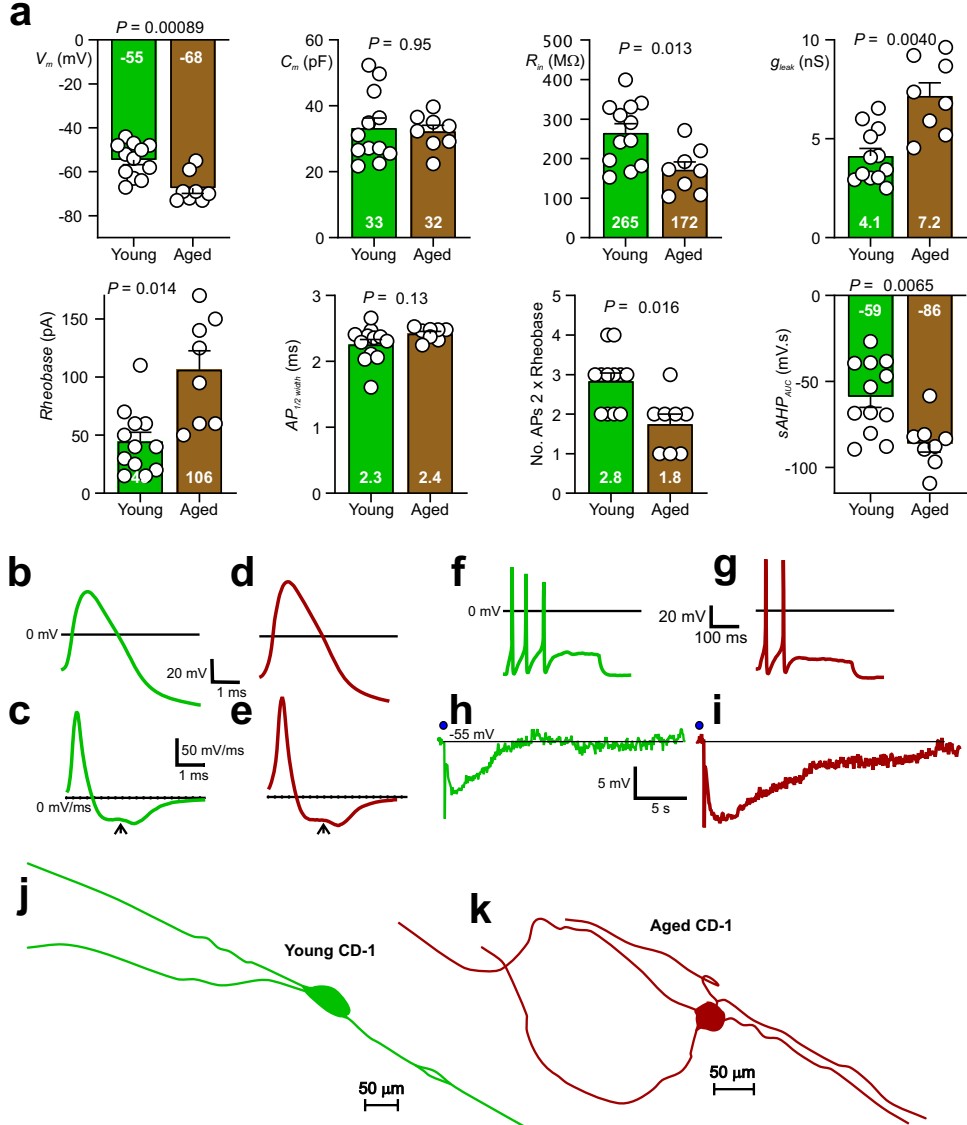

**Fig. 1 Jejunal enteric nervous system intrinsic primary afferent neuron IPAN physiology differs between old and young mice. a** For this and all other bar graphs in the present paper, mean values given at base of bar graphs, open circles represent values measured from individual neurons. Resting membrane potentials ($V_m$) of aged are more polarized than those of young mice. $C_m$ (membrane capacitance = measure of cell surface membrane area) was not different. Input resistance ($R_{in}$), a measure of number of open background channels = reciprocal of cell membrane leakiness)-aged IPANs had lower $R_{in}$. Background leak conductance ($g_{leak}$) was lower for young than for old IPANs. *Rheobase* (threshold current required to evoke one action potential 50% of the time) was lower for young than old IPANs. The width of the action potential at half height of its positive amplitude ($AP_{1/2}$ *width*) was not statistically different for young vs aged IPANs. The number of action potentials evoked at 2 x rheobase intensity (*No. APs 2 x rheobase*) was greater for young compared to aged IPANs. The area under the curve of the postaction potential slow afterhyperpolarisation (*sAHP AUC*) was greater for aged than young IPANs. **b, d** Representative action potential shapes for young and aged IPANs (colour-code: green indicates young, brown indicates aged). **c, e** First order time derivative of action potential showing that both old and young spikes have a hump (arrowhead) on their descending phase, indicating that calcium influx (which contributes significantly to $AP_{1/2}$ width) did not differ notably between young and aged IPANs. **f, g** Example traces of action potentials evoked at 2x rheobase for young and aged IPANs. **h, i** Traces of post-action potential sAHP from young vs aged IPANs. Positions of truncated action potentials indicated by filled circles. **j, k** Examples of IPAN shapes revealed after intracellular neurobiotin dye filling. IPANs had smooth oval cell bodies with multiple long axonal processes running circumferentially within the myenteric plexus (Dogiel type II cell morphology). The full extent of these processes are not shown. Statistics: $N_{young}$ = 12 (from 6 mice), $N_{aged}$ = 8 (from 4 mice). All comparisons made using unpaired t tests except for *No. APs 2 x rheobase* for which a Mann-Whitney test was applied. All tests were two-tailed. All error bars represent the standard error of the mean.

young ones. The rheobase (threshold current for evoking a single action potential (AP) increased 136% from 45 pA ($\eta^2_p = 0.44$), average no. APs evoked by a stimulus pulse injected at twice rheobase intensity decreased by 36% from 2.8 ($\eta^2_p = 0.38$), and the magnitude (area under the curve) of the inhibitory slow afterhyperpolarisation ($sAHP_{AUC}$) evoked by 3 APs increased by 46% from -59 mV.s ($\eta^2_p = 0.35$) (Fig. 1a). Representative traces of

IPAN action potentials are shown in Fig. 1b (young) and Fig. 1d (aged), 1st order time differentials of the APs demonstrated that both young and aged IPANs possessed a hump on the AP descending phase indicating that $Ca^{2+}$ influx contributed to the APs[20,26]. Fig. f & g illustrate AP firing at twice rheobase stimulus intensity for young and aged neurons, respectively. Figure 1h & i show traces of the sAHP for young (h) and aged (i) IPANs. 12

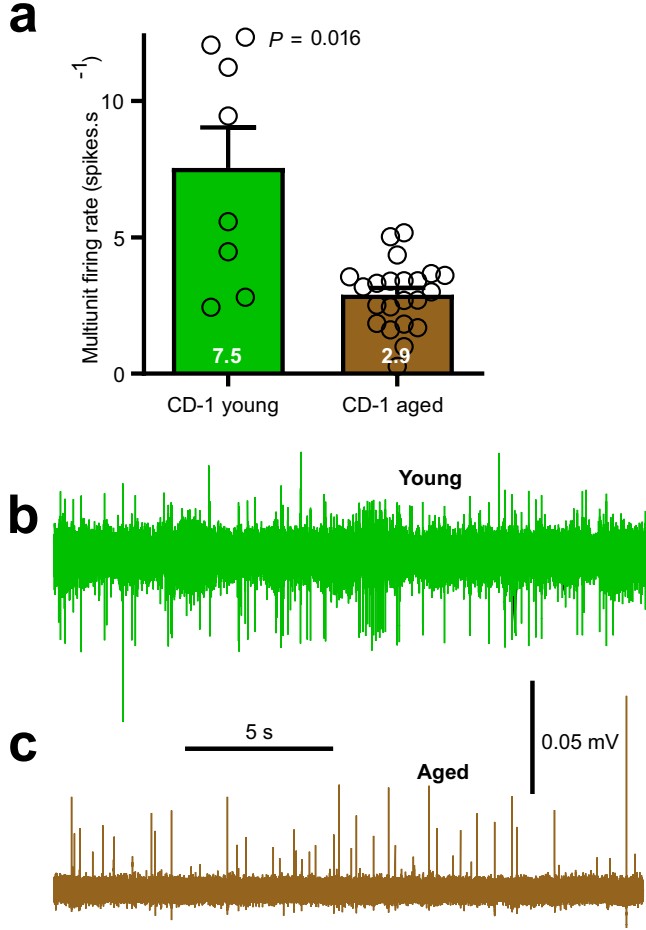

**Fig. 2 Young compared to aged jejunum mesenteric nerve bundle (multiunit) spike firing rates. a** Multiunit spike firing rates were statistically slower for recording made from aged compared to young mice. **b** Representative multiunit trace recorded from young animal. **c** Multiunit trace from aged animal. Statistics: **a** Comparison made using unpaired t test, two-tailed. Nerve fibre bundles: $N_{young} = 8$ and $N_{aged} = 22$. Means given at base of bar graphs, error bars are standard errors.

young and 8 aged IPANs were recorded from, of these 9 young and 7 aged were filed with neurobiotin marker dye and their shape recovered immunohistologically. All 16 had Dogiel type II morphology with smooth round or oval somas and long circumferentially-directed axons. Figure 1j & k show traces of young and aged IPANs revealing typical Dogiel type II morphology. In summary, the sAHP current increased with age. Ageing was also associated with hyperpolarised Vm, increased plasma membrane permeability (decreased $R_{in}$ & increased $g_{leak}$), and a reduction in the number of APs discharged; all of which would be associated with a larger background sAHP current. Contrariwise, cell capacitance, action potential shape and the action potential hump were unaltered by ageing.

**Aged jejunum mesenteric nerve bundles discharge multiunit spikes at a slower rate than those from young mice.** Using a suction recording electrode we recorded the resting mixed (multiunit) AP extracellular spikes discharge from mesenteric nerve bundles attached to short jejunal segments. The average multiunit firing rated was 61% slower for aged compared to young mice (Fig. 2a). Representative examples in Fig. 2 of young vs aged multiunit spikes are given in panel b and c, respectively.

**Aged IPANs exposed to squalamine displayed a young physiological phenotype.** We recorded from aged IPANs (see above) and 30 minutes after completing the first sets of physiological measurements, the Krebs buffer superfusate bathing the myenteric plexus preparation was switched to one that contained 30 μM squalamine lactate[10]; measurements were then repeated. Addition of squalamine had a general excitatory effect increasing the electroresponsiveness of IPANs to resemble that for young IPANs. The onset latency for depolarisation and reduction in $sAHP_{AUC}$ ranged from 4 to 5 min. For passive cell membrane characteristics: $V_m$ depolarized by 21% to -54 mV ($\eta^2_p = 0.61$), $R_{in}$ increased by 65% to 283 MΩ ($\eta^2_p = 0.54$), $g_{leak}$ decreased by 31% to 5.0 nS ($\eta^2_p = 0.32$). For active membrane characteristics: rheobase decreased by 45% to 58 pA ($\eta^2_p = 0.34$), the average number of AP fired at 2x rheobase increased by 67% to 3.0 ($\eta^2_p = 0.47$), and $sAHP_{AUC}$ decreased by 26% to -64 mV.s ($\eta^2_p = 0.29$) (Fig. 3a). $C_m$ and $AP_{1/2width}$ were not altered by squalamine. Representative traces of IPAN action potentials in the presence of squalamine are shown in Fig. 3b-e. Figure 3b shows an AP and 3c gives the 1st order time differentials of the AP confirming that the IPAN AP possessed a hump on its descending phase indicating that $Ca^{2+}$ influx contributed to the AP. Figure 3d & e illustrate the increased AP firing duration at twice rheobase stimulus intensity for an aged neurons in the presence of squalamine (cf. Fig. 1g). Figure 3e depicts the reduced sAHP in the presence of squalamine (cf. Fig. 1i).

**Comparing young to aged vagal single unit resting discharge revealed an ageing code which was eliminated by adding squalamine to the lumen of aged jejunal segments.** Single units were extracted from multiunit vagal nerve recordings using principle component analysis[15]. We measured several distinct parameters that fully describe the firing patterns observed within the 30 min recording periods used for each test sample. The parameters were: mean interspike interval (MII), average burst duration (BD), average gap duration between bursts (GD) and the average intraburst interval (IBI) (Fig. 4a). 70% of single vagal units recorded from young mice discharged ≥ 1 burst. Significantly fewer such events were detected for aged single units (Fig. 4b). When young were compared to aged mice, ageing increased MII, GD, IBI but not BD. MII increased by 427% ($\eta^2_p = 0.20$), GD by 38% ($\eta^2_p = 0.19$) and IBI by 46% ($\eta^2_p = 0.22$) (Fig. 4c).

Addition of 30 μM squalamine to the Krebs buffer perfusing the lumen reduced the sample mean differences for MII, BD, GD and IBI to statistically insignificant 8%, -4%, -5%, and 2%, respectively (Fig. 4d). Plotting fractional changes for aged compared to young mice for each of the 4 parameters revealed a unique, hitherto unknown, ageing code (Fig. 4e). The heat map in Fig. 4f demonstrates that the ageing code is categorically different for those previously calculated for prodepressant (LPS) or antidepressant (JB-1, fluoxetine or sertraline) luminal stimuli acting on the vagus[15]. The ageing code was absent when squalamine was present in the lumen (Fig. 4g) for which the standard error of the mean spanned the zero line for fractional differences.

**The ageing code was conserved across sexes.** The vagal code evoked by young vs. aged CD1 female mice was qualitatively similar for that for male mice (Fig. 5). Thus, both sexes revealed the canonical ageing code of increased MII, increased GD, and increased IBI.

**Swiss Webster mice.** Swiss Webster mice also exhibited the ageing code. We repeated the comparison between 17 young and

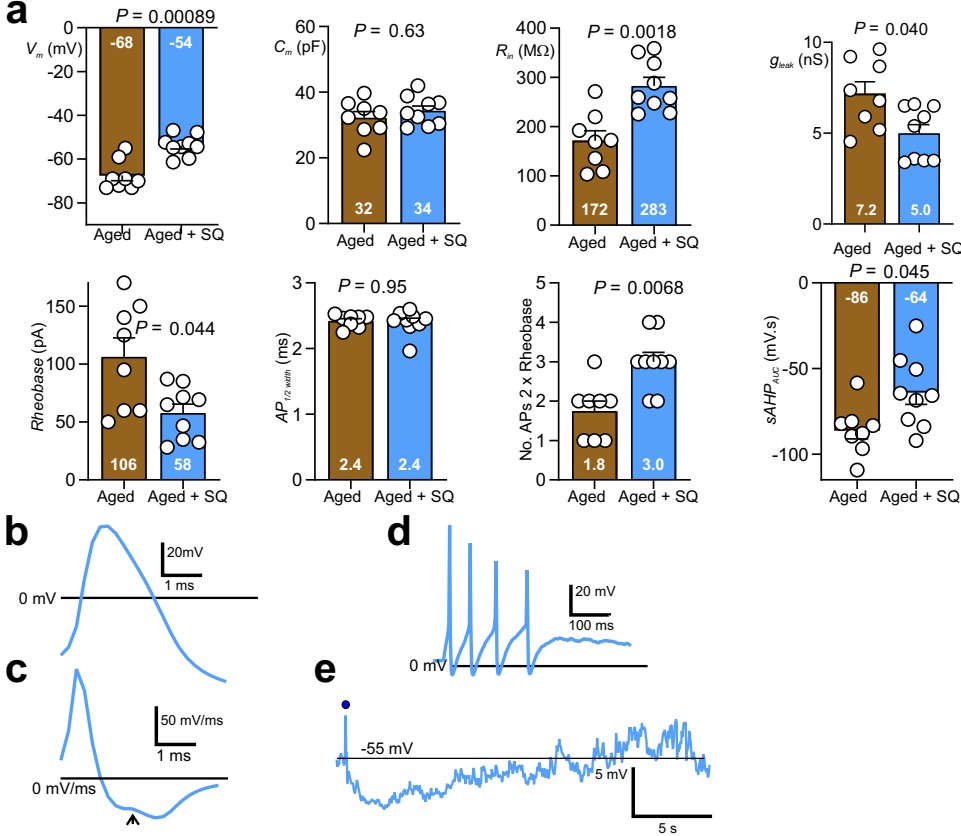

**Fig. 3 Addition of 30 µM squalamine to IPANs changes IPAN physiology from aged to young phenotype. a** Squalamine depolarised $V_m$, increased $R_{in}$, reduced $g_{leak}$ decreased *Rheobase*, increased *No. APs 2 x rheobase*, and decreased *sAHP AUC*. $C_m$ and $AP_{1/2\ width}$ were not statistically different. **b, c** Traces of IPAN action potential (**b**) and first order time derivative (**c**) in the presence of squalamine. **d** Example of IPAN action potential discharge at 2 x rheobase stimulus intensity. **e** Representative post-action potential sAHP. Position of truncated action potential indicated by filled circle. Statistics: All experiments were paired comparing before (luminal Krebs buffer only) and after addition of squalamine to Krebs. $N = 9$ (from 5 mice). All comparisons made using paired t tests except for *No. APs 2 x rheobase* for which a Wilcoxon test was applied. All tests were two-tailed. All error bars represent the standard error of the mean.

12 aged male mice for Swiss Webster (SW) mice and the effects of squalamine on the code to demonstrate that the ageing code is not strain specific. All protocols and the calculation of the ageing code were performed in the same manner as for CD-1 mice.

Clearly, SW mice also exhibited an ageing code when young were compared to aged mice (Supplementary Fig. 1). Additionally, as was the case for CD1 mice, addition of 10 µM intraluminal squalamine altered parameter levels for aged vagal fibres to levels seen in young mice on the ageing code disappeared in the presence of squalamine (Supplementary Fig. 1).

All of 9 young SW and 7 aged IPANs that were iontophoretically injected with Neurobiotin marker dye revealed the multipolar shapes characteristic of IPANs. Examples of SW mouse IPAN shapes revealed after intracellular Neurobiotin dye filling are shown in Supplementary Fig. 2.

**Squalamine but not sertraline augmented afferent aged vagus spike firing.** We have published[15] that for young mice squalamine evokes a vagal code closely resembling that of sertraline. In the current study we compare the effects of 10 µM sertraline with those of 30 µM squalamine[15] on the vagal code. These concentrations were the same as we have used previously[15] for ex vivo vagal single unit recording and were chosen because we had shown that they activate vagal single units by ~20% above resting firing rates for young mice. When this study was

conducted with preparations from aged mice, sertraline increased MII for most single units whereas squalamine decreased MII all units (Fig. 6a), with a greater proportion of single units discharging with ≥1 burst for units exposed to squalamine than to sertraline (Fig. 6b). The changes occurred due to an increase in MII for sertraline (Fig. 6c), and a decrease in MII and IBI for squalamine (Fig. 6d). These data suggest that, unlike for young mice, sertraline and squalamine have opposing effects for aged vagus single unit firing rates. There was also a reduced repertoire for complex discharge patterns involving action potential bursting when sertraline was compared to squalamine.

Representative event markers and sequential rate histograms depicting the 2 different representative single units from aged animals show that while intraluminal squalamine increased (Fig. 7a-c), sertraline decreased the firing rate (Fig. 7d-f). In contrast, sertraline increased the single-unit firing rate for a representative recording taken from a young vagal fibres (Fig. 7g-i). We have previously published that intraluminal squalamine increases the firing rate for young vagal fibres[15]. In register with this increase in young fibre firing rates, mucosal application of 30 µM squalamine (Fig. 7k) or 10 µM sertraline (Fig. 7i) increased the number of IPAN action potentials evoked by intracellular injection of 500 ms duration depolarising current pulses from $2.9 \pm 0.2$ to $5.4 \pm 0.3$ ($P < 0.0001$, paired t test, $N = 11$, 3 mice) for squalamine and from $2.9 \pm 0.2$ to $7.8 \pm 0.2$ ($P < 0.0001$, paired t test for sertraline, $N = 12$, 3 mice).

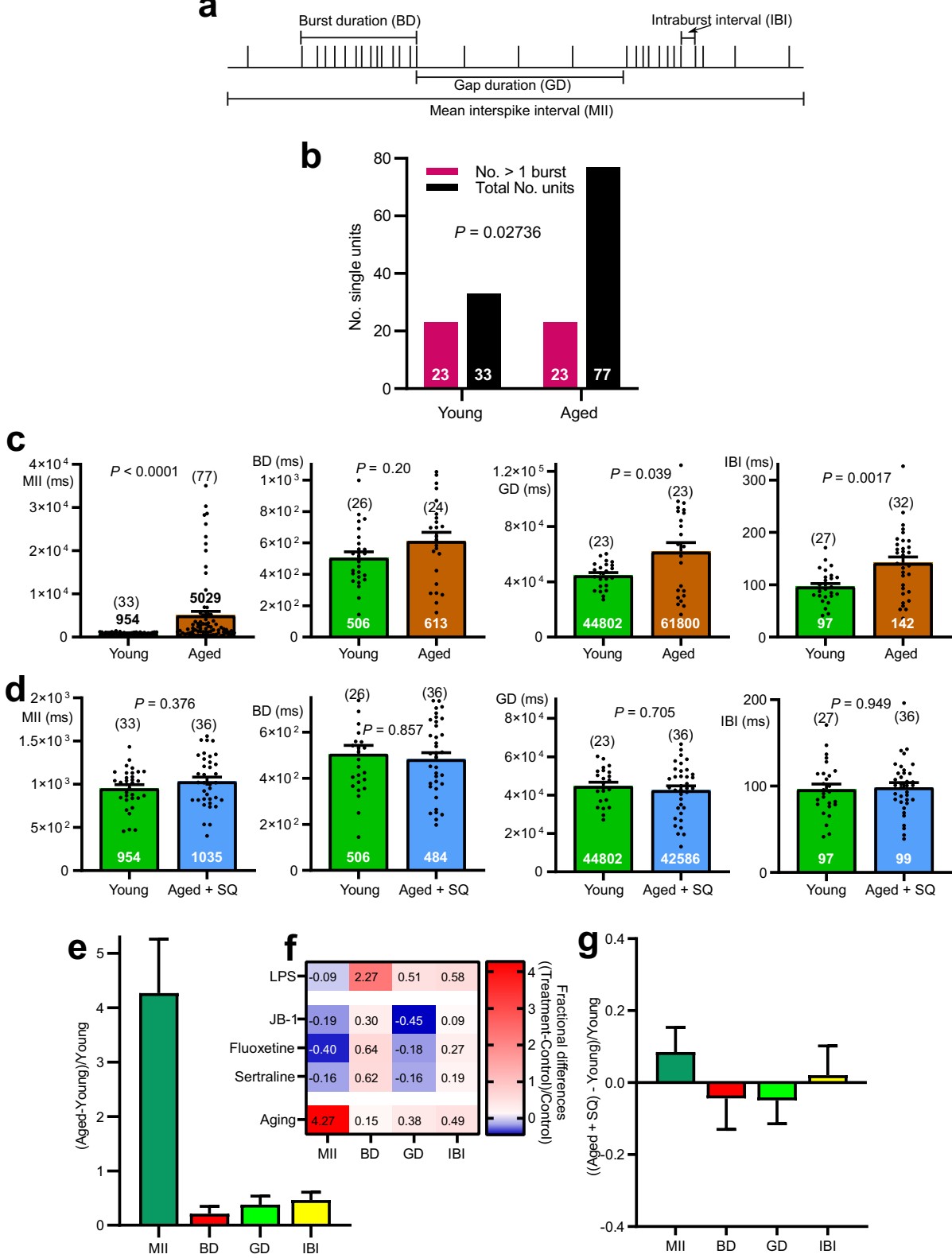

## Discussion

Our findings show IPANs of aged mice differed neurophysiologically from those of young mice. Using ex vivo patch clamp recordings from in situ myenteric IPANs we found that aged IPANs are less excitable than young ones. Aged IPANs had more hyperpolarised $V_m$, a smaller $R_{in}$ and a greater $g_{leak}$. For active membrane characteristics, aged IPANs had a greater *Rheobase*, a smaller No. APs 2 x Rheobase (smaller number of APs discharged at twice *Rheobase* stimulus intensity) and a greater $sAHP_{AUC}$. Addition of squalamine to the Krebs buffer perfusing the gut lumen returned aged IPANs to a young electroresponsive phenotype.

Using extracellular AP recordings, we found that the average firing rate of jejunal mesenteric multiunit spikes was lower for

**Fig. 4 The ageing code, male CD-1. MII, GD, and IBI, but not BD were all significantly increased in aged compared to young single units. a** Diagram of stylised single unit discharge illustrating the 4 parameters measured to quantify individual spike (single unit) firing patterns. The parameters were: mean interspike interval (MII), burst duration (BD), gap duration (GD) and intraburst interval (IBI). **b** Bar graphs showing that the number of vagal fibre single units displaying more than 1 spike burst during the 30 min recording period was greater for young than old mice. **c** Dot plots with superimposed bar graphs (mean ± s.e.m.) showing values for all 4 single unit firing parameters. Nine young mice were compared to 19 aged ones. MII, GD and IBI reached statistical significance. **d** No statistical difference was discernible for any of 4 firing pattern parameters when recordings taken from the 19 young mice luminally perfused with only Krebs buffer were compared to those taken from 10 aged mice whose lumen was perfused with Krebs containing 30 μM squalamine. **e** The ageing code revealed by plotting fractional differences (mean ± s.e.m.) of aged vs. young mice for each of the 4 firing parameters. All of parameters except for BD contributed to the code. **f** Heat map of the parameter fractional differences showing that the ageing code is categorically different from the prodepressant (LPS) or antidepressant (JB-1, fluoxetine, sertraline) codes. **g** The ageing code was eliminated when the lumen of aged animals was perfused with Krebs with added squalamine and compared to young mice whose jejunal lumen was perfused with Krebs only. Statistics: **b** Contingency table analysis by two-sided Fisher's exact test. **c**, **d** Comparisons of parameter means for young vs. aged or young vs. aged + squalamine made using Dunnett's T3 multiple comparisons t tests.

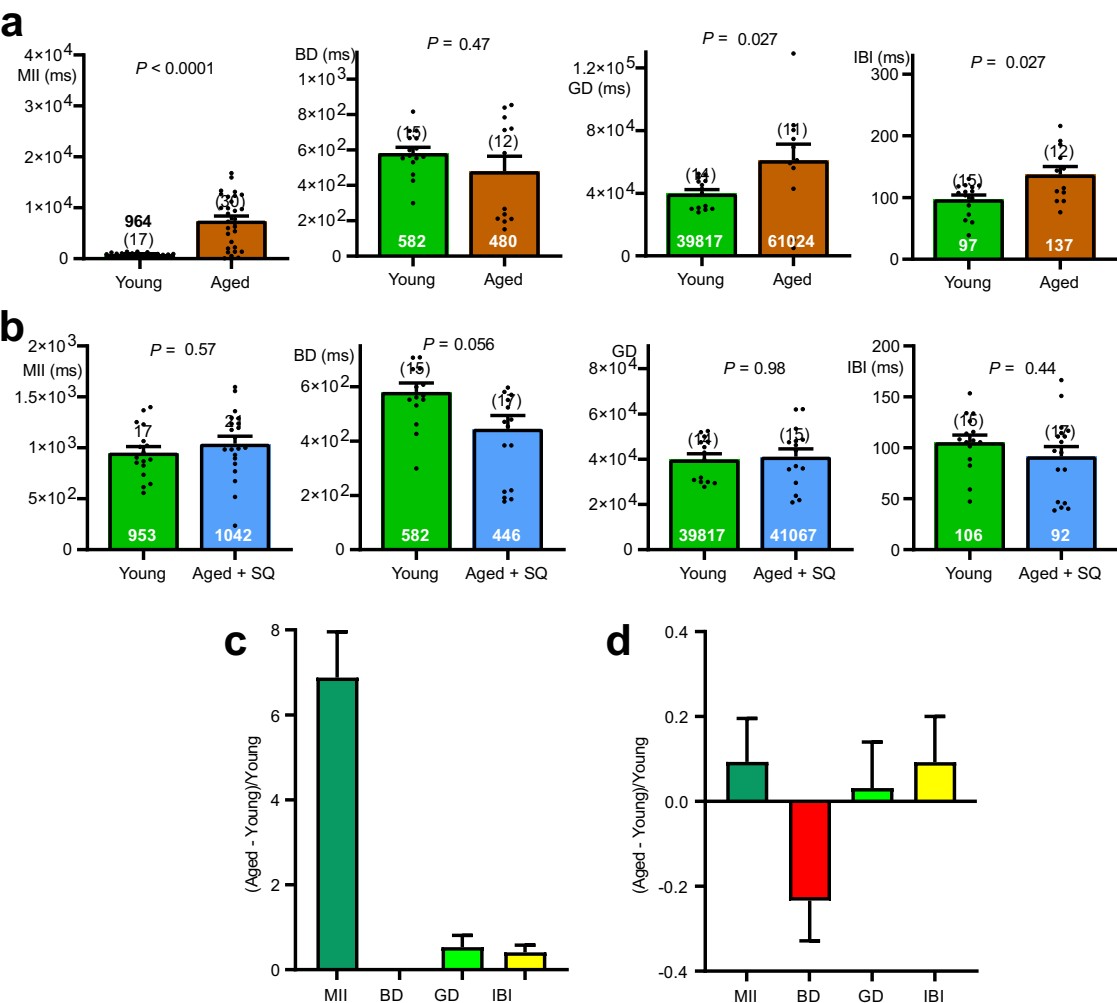

**Fig. 5 The ageing code, female CD-1. MII, GD, and IBI, but not BD were all significantly increased in aged single units. a** Dot plots with superimposed bar graphs (mean ± s.e.m.) showing values for all 4 single unit firing parameters. Five young mice were compared to 8 aged ones. MII, GD and IBI reached statistical significance. **b** No statistical difference was discernible for any of parameters when recordings taken from the 5 young mice perfused with only Krebs buffer were compared to those taken from 5 aged mice whose lumen was perfused with Krebs containing 30 μM squalamine. **c** The ageing code revealed by plotting fractional differences (mean ± s.e.m.) of aged vs young mice. All of parameters except for BD contributed to the code. **d** The ageing code was eliminated when the lumen of aged animals was perfused with squalamine. **c,d**, Comparisons of parameter means for young vs. aged or young vs. aged + squalamine made using Dunnett's T3 multiple comparisons t tests.

aged than for young CD-1 mice confirming and extending a similar previous finding for which only a trend that did not quite reach statistical significance was reported[10].

Single unit recordings revealed that APs recorded from young mouse tissue showed a greater number of bursts than those recorded using aged mice. Aged mice had a larger single unit mean interspike interval (MII), firing gap duration (GD) and within burst intraburst interval (IBI) while burst duration (BD) did not differ statistically. Plotting these parameters as fractional differences aged vs young revealed a unique sensory vagus ageing

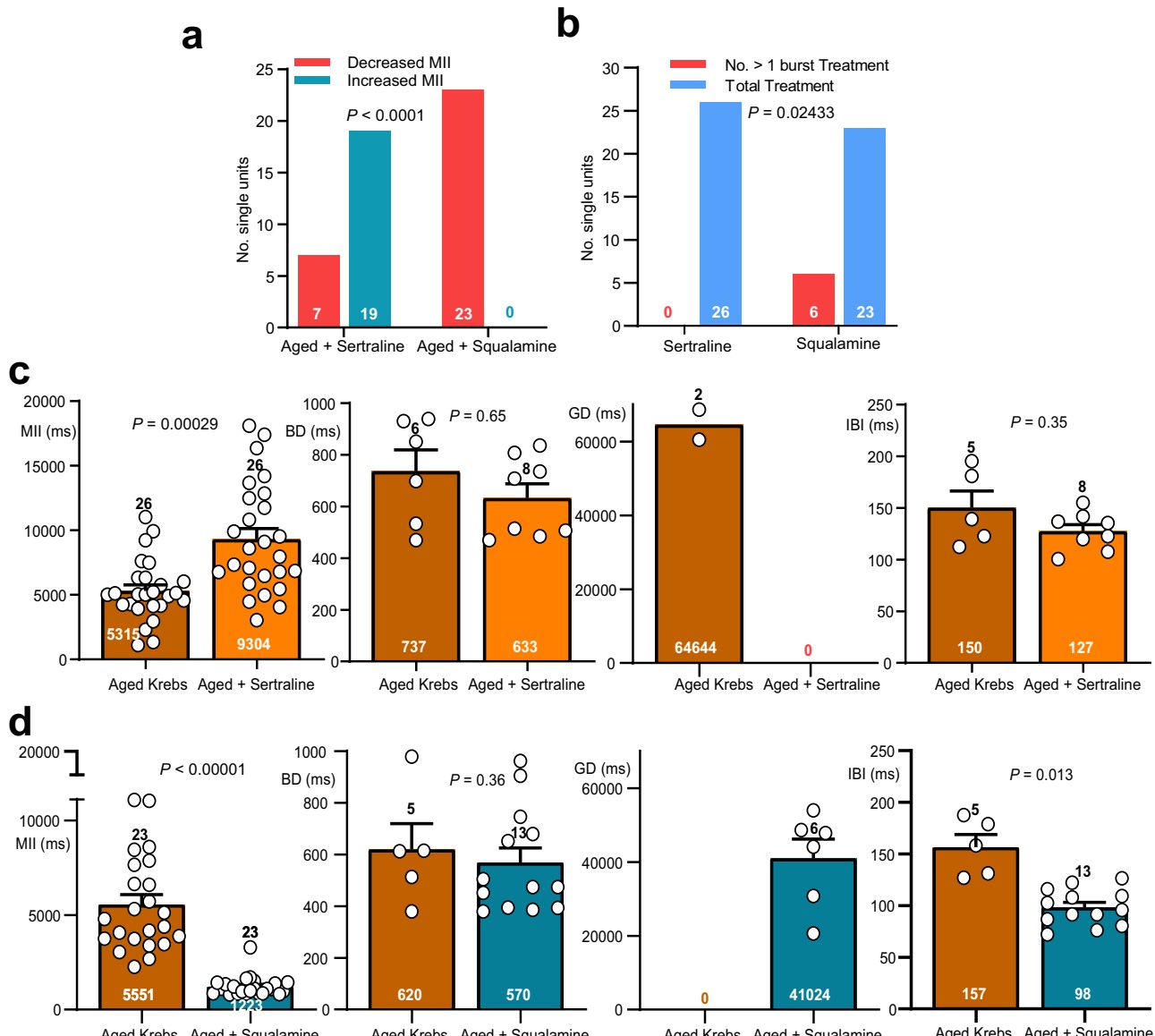

**Fig. 6 Squalamine increased the number of spike bursts and decreased MII & IBI in vagal single units for aged mice; sertraline was ineffective or had opposing effects. a** Squalamine decreased MII for all aged single units tested, sertraline increased MII for more than 70% of units. **b** The proportion of aged single units with more than 1 burst in their firing pattern was higher for units exposed to intraluminal squalamine than for ones exposed to sertraline. **c** Sertraline increased MII for single units from aged mice with no statistically discernable effects on BD, GD or IBI. **d** Squalamine decreased MII and IBI for aged mice. Statistics: **a, b** Contingency table analysis by two-sided Fisher's exact tests. **c, d** Comparisons of parameter means for aged vs. aged + sertraline or aged vs. aged + squalamine made using paired t tests. There were not enough single unit bursts (0 or 1) for aged mice to make statistical comparisons for GD. All error bars represent the standard error of the mean.

code. Addition of squalamine to the superfusate bathing the myenteric plexus abolished the ageing code changing the parameters measured from aged vagal fibres to values seen for young fibres. Interestingly, the ageing code, like the antidepressant code[15], was not confined to only one mouse strain but was also present in Swiss Webster mice.

The vagus-stimulating action of sertraline, but not squalamine, was lost for aged mice. Aged vagal single units exposed to the antidepressant sertraline exhibited fewer bursts than those exposed to squalamine. Sertraline increased MII (slowing the firing rate) while having no statistical effects on the other parameters; in contrast squalamine enhanced vagal firing by decreasing MII and IBI.

Previous studies of ENS and intestinal vagal ageing have been confined only to anatomical changes, however a recent paper

using 8-9 mo old hSNCAA53T constructs as a model for Parkinson's disease showed reduced excitability for IPANs[27]. Colonic migrating motor complexes in aged (24 mo) mice are significantly slower than those in young (3 mo) ones[10] and this is also consistent with decreased function of the aged ENS. Normal ageing has been associated with a plethora of intracellular molecular changes, including abnormal reactive oxygen species and $Ca^{2+}$ levels that trigger mitochondrial dysfunction. In this regard, IPAN mitochondria influence the resting membrane potential and importantly play a role in the post-action potential uptake of $Ca^{2+}$ released from intracellular stores[28]. Indeed, experimental impairment of mitochondria in young adult IPANs has been shown to lead to the exaggeration and prolongation of the sAHP, exactly as has been demonstrated in the present paper for aged IPANs[28].

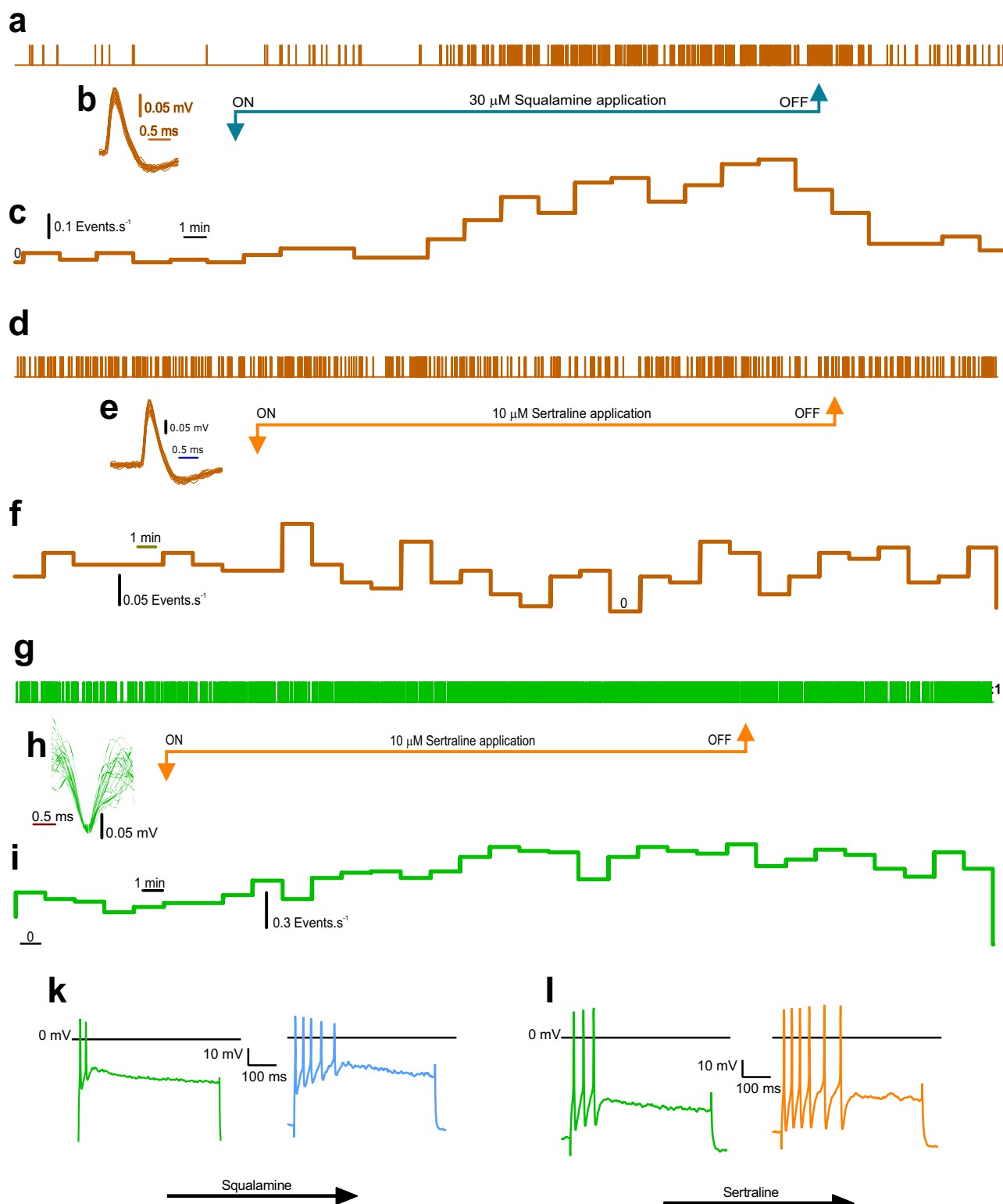

**Fig. 7 Representative action potential event markers of individual single unit responses to intraluminal application of squalamine vs sertraline. a** Event markers showing the occurrence of an individual single unit during its 30 min recording period for intraluminal squalamine test for aged mouse.
**b** Superimposed traces of 26 single units used to generate event markers for **a**. **c** Binned sequential rate histogram showing evolution of excitatory response to squalamine. **d** Single unit event markers showing occurrence of single unit during sertraline test for aged vagus. **e** 25 superimposed traces of single units used to generate event markers for **d**. **f** Binned sequential rate histogram showing reduction in single unit firing in response to intraluminal application of sertraline. **g** Event markers for single unit from young mouse. **h** 20 superimposed traces of single unit used for generation of event markers in **g**.
**i** Histogram showing excitatory response for intraluminal sertraline in young mouse. **k** For young mice mucosal application of 30 μM squalamine increased the number of IPAN action potentials evoked by injection of 500 ms depolarising current pulse at 2x threshold intensity. **l**, Sertraline (10 μM) increased the number of action potentials evoked by injection of a 500 ms depolarising pulse.

We confirmed our earlier results showing a trend that resting CD-1 mesenteric nerve multiunit firing rates were lower for aged than young mice[10], and that intraluminal squalamine could restore the decreased firing rates for aged mice. Our mesenteric multiunit results were also consistent with studies that reported mesenteric nerve multiunit activity was lower for aged (64 y) than younger (47 y) humans and that there were fewer single unit bursts for aged humans.

In the present paper we reveal a canonical ageing code of increased MII, increased GD, and increased IBI for aged vs. young vagal single unit firing patterns. This code was present for both male and female CD-1 mice. Thus, if there are differences across the sexes in afferent vagal signalling, they are likely to be manifest in the central nervous system rather than at the level of the subdiaphragmatic vagus. We had previously published[10] that the single unit mean interspike interval is larger for aged than for young CD-1 mice, and that addition of squalamine to the Krebs buffer perfusing the lumen of the aged jejunum returned MII to values recorded from young mice. However, no neurons in any animal nervous system seem to encode information solely by firing rate, rather firing patterns and intervals must be considered to understand how neurons encode information.

A simpler vagal code analysis has determined that for young (7-12 wks) male rats single unit spike burst frequencies increase in relation to eating[29]. A recent publication[30] using only young 8-16 wk old mice used density-spaced clustering algorithms of spike shapes contained within cervical multiunit signals, recorded in vivo. This method revealed that injected cytokines could be identified by increases in the vagal single unit firing rate that the cytokines elicited[30]. Overall, and to the best of our knowledge, no analogous ageing code to the one being offered in the present paper has yet been published for any nervous system. As we have mentioned in the Introduction, some aged individuals are relatively resistant to the antidepressant effects of SSRIs, but a direct comparison between sertraline and squalamine on the firing rates of aged afferent vagal fibres also has not yet been revealed.

Why does the electrical excitability of IPANs and firing pattern of afferent vagal fibres change with old age? Some insight might be had from our current understanding of the potential mechanisms by which squalamine might be acting. We have previously shown that local administration of squalamine restores electrical activity in IPANs from Parkinson's disease mouse models genetically engineered to accumulate aggregates of alpha-synuclein within the enteric neurons[27]. Normal peristalsis was restored and brain-directed vagal afferent is stimulated. In addition, orally administered squalamine successfully restored gut motility and several neurological symptoms in elderly patients with Parkinson's disease-associated constipation in two a recently completed Phase 2b clinical trials[31,32], demonstrating the translatability of the preclinical observations to humans.

In aqueous solution squalamine exists as a zwitterion with a net positive charge. As a consequence of its chemical structure it is highly amphiphilic, it is both highly water soluble and membrane active, and will bind electrostatically to membranes that contain anionic phospholipids and subsequently embed within the membrane[33–39]. Furthermore, previous studies have demonstrated that squalamine can effectively both displace proteins that are bound electrostatically to neuronal membranes and additionally, prevent their initial aggregation on the membrane surface[40]. For example, studies in C. elegans, engineered to express an aggregating human mutation in alpha-synuclein, develop paralysis as aggregates of alpha-synuclein accumulate within their excitable muscle cells[40]. Exposure of these worms to increasing concentrations of squalamine results in a proportional reduction in the number of protein aggregates and a dose-dependent increase in motility[40]. Recent studies have demonstrated that misfolded proteins, defined as those that resist proteinase digestion, accumulate with ageing in all organs of the mouse[41]. These ageing-associated misfolded proteins could include alpha-synuclein, since numerous studies have reported that alpha-synuclein increases with age in older rats, monkeys, and humans[42–45]. These data are not surprising, as age is a key risk factor for many neurodegenerative disorders. Based on these observations we speculate that squalamine might improve the electrical excitability of the IPANs in the aged mouse through displacement of misfolded proteins from cellular membranes involved in neuronal electrical activity.

Numerous studies have shown that once integrated into a membrane the spatial organization of lipids within the membrane, fluidity, and tensile strength are altered[37]. Because squalamine electrostatically reduces the overall surface charge of the membrane, the function of membrane proteins positioned by electrostatic forces can be affected. For example, the application of squalamine to a mouse cortical neuron ex vivo activates the synaptic AMPA receptor[46], and in other experimental settings inhibits the Type 3 sodium hydrogen exchanger, which regulates intracellular pH[47]. Thus, it is also possible that squalamine could enhance electrical excitability of the aged IPAN as a consequence of the modulation of the activity of membrane-associated proteins. The precise mechanism by which squalamine restores the electrical activity of the aged IPAN to a more youthful phenotype, however, remains to be determined.

Squalamine does have antibiotic activity and could perhaps alter gut propulsive activity either directly by stimulating IPANs (as shown here) or by lessening of Parkinsonian intestinal dysbiosis[48,49]. However, antibiotics may have conflicting effects on gut motility: bacitracin, neomycin, and penicillin V increase colonic propulsion[50,51] while vancomycin or ampicillin decrease faecal output[52]. It is not clear that squalamine's antimicrobial activity, per se, can explain its therapeutic effects on constipation in Parkinson's disease.

In conclusion, our findings show that ageing is associated with decreased excitability of intrinsic primary afferent neurons in the enteric nervous system and a specific afferent jejunal gut-brain axis ageing code deduced from the resting firing pattern of vagal single unit fibres. The ageing code could be suppressed by intraluminal squalamine. We also showed that whilst sertraline decreased the firing rate of vagal afferent single units, squalamine retained the ability to excite vagal fibres for the aged vagus. These findings suggest targeting the vagus nerve using pharmacological or electroceutical approaches may be a productive research area for age related disorders involving the gut-brain axis.

## Methods

**Animals**. Six to eight-week-old female and male CD-1 or male Swiss Webster (SW) mice were purchased from Charles River (Montreal, QC, Canada). Young animals (3 mo) were allowed to habituate to the animal facility for 1 week while older animals were housed until they were 18–24 mo of age when they were used for experimentation. Animals were housed 4/cage and under controlled conditions (21∘C) on a 12-h light/dark cycle (lights on at 5:00 a.m.) and fed ad libitum. All experiments were carried out in accordance with the guidelines of the Canadian Council on Animal Care and ARRIVE Guidelines and were approved by McMaster University's Animal Research Ethics Board (Animal Utilisation Protocols: 16-08-30 & 20-05-21). Mice were euthanized by cervical dislocation and all action potential recordings performed ex vivo.

**Enteric nervous system**. A 2 cm segment of ileum was removed from freshly euthanized mice and the tissue was placed in a 2 ml

recording petri dish whose inside base was lined with sylgard (170 silicone elastomer, Dow Corning, Midland, MI, USA) and filled with Krebs buffer of the following composition (mM): NaCl 118.1, KCl 4.8, NaHCO$_3$ 25, NaH$_2$PO$_4$ 1.0, MgSO$_4$ 1.2, glucose 11.1, CaCl$_2$ 2.5; the buffer was continuously bubbled with carbogen (95% O$_2$–5% CO$_2$). Nicardipine (3 μM) (Sigma-Aldrich, Oakville, ON, Canada) was routinely added to the saline to prevent spontaneous muscle contraction. The segment was opened along a line parallel to the mesenteric attachment and pinned flat, under moderate tension, mucosa uppermost. The myenteric plexus was exposed by dissecting away the mucosa, submucosa, and circular muscle. The recording dish was then mounted on an inverted microscope (Nikon Eclipse TE 2000-S,Melville, NY, USA) and imaged via a PC computer using a Rolera-XR camera (Surrey, BC, Canada) and the tissue continuously superfused (4 ml min$^{-1}$) with carbogenated Krebs warmed to 34 °C. A ganglion was prepared for patch clamping as described previously[53]; briefly, the selected ganglion was exposed by gravity flow for 15 min to 3 ml of 0.02% protease type XIV in Krebs saline (Sigma-Aldrich), then the upper surfaces of myenteric neurons were revealed by cleaning part of the ganglion with a fine hair until individual neuron soma became just visible. As noted previously[53] there was no evidence of cell swelling after this gentle treatment.

Signals were measured in voltage recording (current clamp) mode using an Axon Instruments Multiclamp 700 A computer amplifier (Molecular Devices, San Jose, CA, USA), and a Digidata 1322 A (Axon Instruments) digitizer was used for A/D conversion. Signals were low pass, 4-point Bessel filtered at 5 kHz, and then digitized at 20 kHz. Data were stored on computer and analyzed offline using Pclamp software (Molecular Devices). Voltage or current commands were delivered to the amplifier under computer control using Clampex 9 (Molecular Devices) software. Patch pipettes were pulled on a Flaming-Brown-P97 (Sutter Instrument, Novato, CA, USA) electrode puller to produce micropipettes with resistances 4–6 MΩ. The patch pipettes were made from thick-walled borosilicate glass (Sutter Instrument) and filled with a solution of the following composition in mM: KMeSO$_4$ 110-115, NaCl 9, CaCl$_2$ 0.09, MgCl$_2$ 1.0, HEPES 10, Na$_3$GTP 0.2, BAPTA.K$_4$ 0.2 with 0.2 % neurobiotin (Vector Laboratories, Newark, CA, US) 14 mL KOH to bring the pH to 7.3. The online program Maxchelator (Maxchelator:https://somapp.ucdmc.ucdavis.edu/pharmacology/bers/maxchelator) gives a predicted value free [Ca$^{2+}$] of 0.18 μM at 34 °C[54] for this intracellular solution. This value is close to resting free [Ca$^{2+}$] as estimated using Ca$^{2+}$-sensitive dyes in guinea pig Dogiel type II neurons (IPANs)[55,56].

With the amplifier in voltage-clamp recording, about 50 hPa positive pressure was internally applied to the pipette before its tip entered the Krebs buffer superfusing the myenteric plexus preparation; the pressure was maintained until the tip was in close apposition to a neuron membrane. Only recordings with seal resistances ≥ 4 GΩ were used for analysis. After gigaseal formation, the amplifier was switched to current clamp recording and whole cell recording mode entered by further suction. During the recording period, depolarising or hyperpolarising current pulses could be injected, under computer control, via the patch pipette using Pclamp 9 Clampex software (Molecular Devices). Access resistance and cell membrane resistance, capacitance and time constants, were periodically monitored by software programmed switching to the Pclamp membrane test protocol which injects square wave pulses oscillating about the holding potential.

At the end of each recording, neurons were ionophoretically loaded with neurobiotin by passing 40 × 500 ms duration +0.1 nA current pulses via the patch pipette. The tissue was fixed in Zamboni's fixative (2% v/v picric acid, 4% paraformaldehyde in 0.1 M Na$_2$HPO$_4$/NaH$_2$PO$_4$ buffer, pH = 7.0) overnight at 4 °C, and then cleared using 3 ×10 min washes of DMSO followed by 3 × 10 min washes with PBS. The tissue was then exposed to streptavidin-Texas Red (Vector Laboratories), diluted 1:50, to reveal neurobiotin. After final rinsing, the tissue was mounted in PBS containing 80% glycerol and 0.1% NaN$_3$ and viewed under fluorescence epi-illumination on an inverted microscope (Nikon Eclipse TE 2000-S,Melville, NY, USA) and imaged using a Rolera-XR camera (Surrey, BC, Canada) Texas Red (596 nm & 620 nm excitation and emission peaks). Shapes of fluorescing neurons were traced using Inkscape 1.2 (Inkscape Project, available from: https://inkscape.org).

**Mesenteric nerve recording.** Mice were sacrificed by cervical dislocation and all action potential recordings performed ex vivo. Short (2.5 cm) segments of proximal jejunum with attached mesenteric arcade containing a single neuromuscular bundle were immediately removed and placed in a sylgard lined recording petri dish filled with Krebs buffer. The segment was emptied of contents using a syringe filled with Krebs, then both ends were cannulated with silicone tubing. The gut and mesenteric tissue were pinned to the sylgard using pins cut from 0.25 mm diameter tungsten wire and the mesenteric nerve bundle exposed by microdissection under a stereomicroscope. The preparation was then transferred to a Nikon Eclipse TE 2000 inverted microscope and the lumen gravity perfused at 1 ml/min with room temperature (22 °C) carbogenated Krebs or Krebs plus 30 μM squalamine dilactate using several Mariotte bottles[57] attached to a plastic manifold. The serosal compartment was separately perfused at 5 ml/min with prewarmed (34°C) Krebs solution to which 3 μM nicardipine had been added to isolate vagal chemosensory responses by preventing active muscle contractions but not vagal responses to distension[58].

The cleaned nerve was sucked into a glass-recording pipette attached to a patch-clamp electrode holder, and extracellular nerve recordings made with pClamp software using a Multi-Clamp 700B amplifier and Digidata 1440 A signal converter (Molecular Devices). The nerve bundle within the pipette was isolated from the Krebs within the recording dish by gently pressing the tip into fat tissue adherent to the uncleaned parts mesenteric arcade. Electrical signals were bandpass-filtered at 0.1–2 kHz, sampled at 20 kHz, and displayed and stored on a personal computer[58].

Baseline recording with Krebs buffer in the gut lumen was performed for 15 min to verify that the resting firing rate was stationary using Pclamp software; samples with non-stationary discharge (windup or rundown) were discarded. Then recording continued for 30 min which constituted the test period for young vs aged comparisons. For experiments where squalamine or sertraline were added to the luminal perfusate, the Krebs buffer only control recording period was 30 min and this was followed by another 30 min of recording in the presence of either drug. Rundown of constitutive vagal discharge in this system is not evident until >90 min of recording[58]. Only one luminal test additive was applied once per animal to avoid possible signal rundown. After recording responses to luminal test substances and to allow post-hoc identification of vagal single units, we applied 0.2 ml CCK to the serosal surface of the jejunum using a handheld micropipette. Finally we distended the intestine by raising the intraluminal pressure to 14 hPa to demonstrate that the isolated single units could still respond to distension. Testing for the response of each of the isolated single units to CCK is a well-established method for identifying vagal fibres within the mesenteric nerve bundle[58,59]. Cholecystokinin (25–33) sulphated

(AnaSpec, Fremont, CA, USA) was dissolved in dimethyl sulfoxide (DMSO) to make a 1 mM stock solution. Aliquots were diluted on the day of the experiment to a working concentration of 0.1 µM in Krebs buffer, with a final DMSO concentration ≤0.0001%.

We tested the following psychoactive agents: 10 µM sertraline hydrochloride[15] (MilliporeSigma, Burlington, MA, USA). Squalamine dilactate was provided by Dr Michael Zasloff, Georgetown University (Washington, DC, United States). Squalamine dilactate powder was dissolved in 90% ethanol to make a stock solution, then aliquoted and stored at -20 °C until use. Stock solution was diluted in Krebs buffer to a working concentration of 30 µM for in vitro experiments[15]. These concentrations activate young adult vagal fibres by approximately the same intensity of ≈20% above baseline firing rates.

**Analysis of single-unit firing patterns**. Each analysed vagal single unit was discriminated from others in the multiunit recording using principal component analysis of their action potential shape, amplitude and width using the Dataview program[60] for extracellular action potential analysis. Single units belonging to each vagal axon were converted into a single event point processes and displayed and used for further analysis[15].

For each single unit event channel in Dataview point processes intervals vs time were displayed as using the event parameter histogram plot option and the mean interspike interval (MII) read from the descriptive statistics panel.

Event bursts were detected by the Poisson surprise method[15,61]. For each control or treatment event channel being measured in Dataview the Event analyse: Histograms/statistics option of the programme calculated the gap (GD) and burst (BD) durations[15]. For intraburst intervals (IBI) and the Krebs and treatment bursts event channels that were created by the Poisson surprise method were logically combined using the AND gate function, thus extracting only the bursts from the point process events for either the control or treatment recording periods.

**Statistics and reproducibility**. Descriptive statistics were calculated in GraphPad Prism ver. 8.3 (GraphPad Software, San Diego, USA) are given as mean ± standard errors. When a statistical test was performed, the P value given is the probability of the test statistic being at least as extreme as the one observed if the null hypothesis of no difference is admitted. The partial eta squared statistic $\eta^2_p$[62, pp. 70-71] gives the effect size for differences calculated in the t-test module within GraphPad. According to Cohen's guidelines[63] for interpreting $\eta^2_p$, 0.01 indicates a small, 0.06 a medium and 0.14 a large effect size. Fractional changes in measured parameters, each given as mean ± standard error, were performed in GraphPad, which also calculated the propagated standard error for the fractional changes.

**Reporting summary**. Further information on research design is available in the Nature Portfolio Reporting Summary linked to this article.

## Data availability
The authors declare that the data supporting the findings of this study are available within the paper and its supplementary information files. Source data underlying figures are provided in Supplementary Data 1.

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

## Acknowledgements

This work was supported by grants RGPIN-2019-05982 & RGPIN-2021-03816 from the Natural Sciences and Engineering Research Council of Canada Discovery Grant Program awarded to WK. It was also supported by a Clifton W. Sherman Scholarship and a Queen Elizabeth II Graduate Scholarship in Science & Technology to CW.

## Author contributions

K.M.N., C.W., Y.M. performed the experiments. W.K. designed the study. and wrote the initial manuscript draft. K.M., A.S., D.B. & M.Z. contributed to study design and helped with the initial draft. P.F., M.A., H.H., E.I. helped with revisions and drafting. This paper is dedicated to the memory of John Bienenstock, Distinguished University Professor of Pathology and Molecular Medicine at McMaster University. John was the Founding Director of the Brain-Body Institute and a mentor to innumerable graduate students, post-doctoral fellows and visiting scientists. He was a personal friend and the inspiration behind the work described in the present paper.

## Competing interests

The authors declare no competing interests.
