## [Peer Review File · Communications Biology]

Reviewers' comments:

Reviewer #1 (Remarks to the Author):

The manuscript by Neufeld et. al. describes the neuro-electrophysiology of myenteric intrinsic primary afferent neurons (IPAN) preparations from the intestinal segments of aged and young mice. The neurons were characterized for various electrophysiological patterns (aging code) to record firing patterns such as resting membrane potential, input resistance, leak conductance, membrane capacitance, excitability and afterhyperpolarisation. The major conclusion from the study is that aged neurons presented decreased firing patterns as compared to the neurons from young mice, and treatment with squalamine can reverse the electrophysiological patterns of aged neurons. The study is adequately planned, the results are interesting and the overall conclusion moves the field forward. The manuscript is suitable for publication in *Communication Biology*, however, there are some concerns that need to be addressed before any form of publication.

1. Reference no 10 is a publication from the same team that described the vagal firing pattern in relation to the reduced colonic motility in aged vs young mice where squalamine was able to reverse the aged-based reduction in motility. It is important to stand out the current study from this previous publication by describing the differences. It looks like the previous study was more focused on gastric motility/firing rate and the current paper is using the same concept but more electrophysiological details of neuronal firing patterns. It could clearly undermine the novelty of the current paper.
2. The parameters of V_m , R_{in} and I_{leak} are deranged in the aged mice in comparison to the young (Fig 1). The underlying mechanism of this aging can be attributed to aging/reduction in ion channels on the membrane. The hump in Fig1 C and E, indicating the role of Ca channels, looks similar in aged and young neurons. Shouldn't it be different in aged and young mice, considering the ion channel-related parameters are being disturbed with aging?
3. In figure 3, the introduction of squalamine in the buffer restored the aging-associated firing parameters (phenotype) of the IPANs. How it can be differentiated as rejuvenation/aging reversal vs hyperexcitability of the neurons? Any intracellular changes, protein up/down regulations or expression patterns or any discussion that can describe it as more than just restoring phenotype.
4. The authors proposed a mechanism for squalamine that it can displace the aggregated and misfolded proteins from the cell membrane. Can it be confirmed by imaging those cell membranes for misfolded protein aggregates before and after treatment with squalamine?

Reviewer #2 (Remarks to the Author):

In this study conducted by Neufeld et al., the researchers investigated the mixed extracellular action potentials of ileum myenteric intrinsic primary afferent neuron (IPAN) and the extracellular discharge from individual single vagal fibres in jejunal mesenteric nerve bundles. A comparison of firing patterns between old and young animals was made. Surprisingly, the authors discovered that infusing jejunal luminal squalamine could restore the vagal firing rates and responses to selective serotonin reuptake inhibitors (SSRIs) in aged mice. This finding is particularly interesting in the context of Parkinson's disease research because Squalamine is known to displace membrane-bound aggregates of misfolded alpha-synuclein. The study is well-designed and provides sufficient data.

One minor concern is that the authors should also discuss the broad-spectrum antimicrobial activity of Squalamine in the context of Parkinson's disease-associated constipation.

Reviewer #3 (Remarks to the Author):

The authors describe differences in firing pattern and cell membrane characteristics of young- and old-derived neurons from the enteric and vagal systems.

General comments:

Although the results are somewhat expected (that young- and old-derived neurons show differences), it is important to have a deep characterization of these differences. So I think the article is of interest, however significant ameliorations are needed.

- The n of mice used; and not only fibers analyzed; is of extreme importance, and not cited. Statistical analysis should consider the number of mice used.
- The use of only males is not in line with "good practices" and guidelines in science (<https://orwh.od.nih.gov/sex-gender/nih-policy-sex-biological-variable>). Therefore I strongly recommend that the authors include females in their study or at least strongly justify why the use of only males. Adding females to the study and reporting on potential differences could be of great interest given the many sex-bias disease related to age and gut-brain axis signaling (e.g. Alzheimer's, Parkinson's...).
- The text should be revised for mistakes and to improve "fluidity". As it is now this article is hard to read.
- It would be interesting to see how the young-derived neurons react to the drugs used to treat old-derived neurons.
- More details about the drugs used in the introduction would be of interest. How does this drug works? Why the authors chose that ones?
- figure presentation should be improved; colors and consistency (whenever possible make the graph in the same style).
- the conclusion contains some over-statements, same for the title which is somewhat misleading.

Specific questions/comments:

- why only jejunum neurons were analyzed?
- Fig 1 not clear from which area are the neurons analyzed.
- Fig 2 and Sup 1a why this difference in young and old analyzed? It seems like a big difference in group sizes that may result in statistical errors. for Fig1, a dot-plot as done in other figures should be shown here.
- Fig 4g and Sup 1d; why there is this "cut" in the graph bar if there is nothing on the top part of the graph?
- Fig 5c; not clear/described what are the number un the bottom of the columns of the graphs. The number for the dot is useless and only pollute the graph.

Comments on the text:

Line 32 - the word reverted is a too strong of statement as some aspects of the aged neurons are not reverted with the drug.

Line 37 - examples of how these results may have implications in treatments should be added.

Lines 78 to 83 - Sentences hard to read and not clear; should be either removed or improved and examples should be added, e.g how could we develop a model of aged ENS?

Lines 101, 103 - ref missing

Lines 484 to 488 - "over conclusion" remove - no evidence is provided to support the idea of anti-aging effects that can determine the state of the gut microbiota.

Dear Dr. Kunze,

Your manuscript entitled "Gut-Brain rejuvenation: Identification of age associated changes in the neural firing pattern of myenteric primary afferent neurons and vagal fibres that are reversed by the aminosterol, squalamine" has now been seen by 3 referees, whose comments are appended below. You will see from their comments copied below that while they find your work of potential interest, they have raised quite substantial concerns that must be addressed. In light of these comments, we cannot accept the manuscript for publication, but would be interested in considering a revised version that addresses these serious concerns.

We hope you will find the referees' comments useful as you decide how to proceed. Should further experimental data or analysis allow you to address these criticisms, we would be happy to look at a substantially revised manuscript. However, please bear in mind that we will be reluctant to approach the referees again in the absence of major revisions.

In particular, we ask that you:

- (1) Please address the discrepancies between aged and young neurons mentioned by Reviewer #1 (point 2). It would also be important include additional mechanistic data to evaluate points #3-4 from Reviewer #1, to substantiate your discussion of how squalamine might impact neuronal activity. We would also encourage you to include additional data on how young-derived neurons respond to pharmacological interventions, as mentioned by Reviewer #3.
- (2) While we would not require that you repeat all experiments using female samples (as mentioned by Reviewer #2), a revised manuscript should clearly justify the use of male samples, and mention this point as a limitation of the study.
- (3) Please carefully address the discussion points raised by each reviewer, and qualify the conclusions of the study as best outlined by Reviewer #3. Please also clearly distinguish the novelty provided by this study over your previous work in reference #10.

Please do not hesitate to contact me if you have any questions or would like to discuss the required revisions further. Thank you for the opportunity to review your work.

Sincerely,

George Inglis

George Inglis, PhD
Senior Editor
Communications Biology
orcid.org/0000-0002-9069-5242

Referee expertise:

Referee #1: Gut-brain axis, pharmacological interventions

Referee #2: Gut-brain axis

Referee #3: Gut-brain axis

Reviewers' comments:

Reviewer #1 (Remarks to the Author):

The manuscript by Neufeld et. al. describes the neuro-electrophysiology of myenteric intrinsic primary afferent neurons (IPAN) preparations from the intestinal segments of aged and young mice. The neurons were characterized for various electrophysiological patterns (aging code) to record firing patterns such as resting membrane potential, input resistance, leak conductance, membrane capacitance, excitability and afterhyperpolarisation. The major conclusion from the study is that aged neurons presented decreased firing patterns as compared to the neurons from young mice, and treatment with squalamine can reverse the electrophysiological patterns of aged neurons. The study is adequately planned, the results are interesting and the overall conclusion moves the field forward. The manuscript is suitable for publication in *Communication Biology*, however, there are some concerns that need to be addressed before any form of publication.

1. Reference no 10 is a publication from the same team that described the vagal firing pattern in relation to the reduced colonic motility in aged vs young mice where squalamine was able to reverse the aged-based reduction in motility. It is important to stand out the current study from this previous publication by describing the differences. It looks like the previous study was more focused on gastric motility/firing rate and the current paper is using the same concept but more electrophysiological details of neuronal firing patterns. It could clearly undermine the novelty of the current paper.

Response:

Although it might have been expected that aged vs young neurons (vagal & myenteric) differ neurophysiologically, there could be no a priori knowledge about the nature of the difference. As was mentioned in our Introduction much had been published about relevant anatomical differences, also our previous paper¹ did show that the average vagal firing rate was reduced in aged vs young mice and that motility was reduced. But a single parameter such as vagal firing rate and a change in motility is in no way the same concept as a canonical vagal aging code for single fibre firing patterns.

Moreover, a reduction in vagal firing rate is not necessarily a marker code for aged vagal function because the antidepressant bupropion slows vagal single unit firing in young mice² and it is unlikely that this antidepressant, used in adult humans, alters vagal function to resemble that seen in aged animals. It is for this reason that an underlying, and previously undescribed, ageing code was suspected, and our experimental results are consistent with this.

2. The parameters of V_m , R_{in} and g_{leak} are deranged in the aged mice in comparison to the young (Fig 1). The underlying mechanism of this aging can be attributed to aging/reduction in ion channels on the membrane. The hump in Fig1 C and E, indicating the role of Ca channels, looks similar in aged and young neurons. Shouldn't it be different in aged and young mice, considering the ion channel-related parameters are being disturbed with aging?

Response:

The reviewer raises an interesting point. Since no electrophysiological recordings have yet been published for aged enteric neurons, it might perhaps be assumed that all ion channel conductances would be altered by aging. There is no extant theory to predict that the action potential Ca^{2+} hump might be altered during aging. The hump is nevertheless reduced or abolished in young adults during intestinal Nematode infection and inflammation³; but this is not the same as aging. Even after many decades of research, the ion channel conductances involved and their manner of intracellular coupling in generating the IPAN action potential slow afterhyperpolarisation (sAHP) has not yet been fully elucidated, even for young adult animals. The hump on the descending phase of the action potential is mainly generated by an inward Ca^{2+} current. In contrast, it is known that the Ca^{2+} ions directly generating the slow afterhyperpolarisation (sAHP) derive from intracellular sources with complex interplay between endoplasmic reticulum and mitochondrial Ca^{2+} stores⁴. In summary, there is no *a priori* reason to expect an alteration in the action potential hump in healthy aged animals.

3. In figure 3, the introduction of squalamine in the buffer restored the aging-associated firing parameters (phenotype) of the IPANs. How it can be differentiated as rejuvenation/aging reversal vs hyperexcitability of the neurons? Any intracellular changes, protein up/down regulations or expression patterns or any discussion that can describe it as more than just restoring phenotype.

Response:

“Hyper-“meaning excessive is not easy to fit into our description of the actions of squalamine. We could say that the electroresponsiveness elicited by squalamine for aged IPANs mirrored that of young IPANs, a functional “rejuvenation”, but we did not intend to suggest this was aging reversal. Hyperexcitability in enteric neurons, is usually related to gastrointestinal inflammation/infection of the enteric nervous system. For example, IPANs in the presence of trinitrobenzene sulphonate induced inflammation exhibit spontaneous action potential firing⁵. Similarly, inflammation due to nematode infection results in the spontaneous generation of action potentials³. In contrast, application of squalamine in our experiments yielded a complete absence of spontaneous action potentials, Normal numbers of IPAN action potentials discharged by suprathreshold (but less than 2x threshold) 500 ms pulses have been reported^{3, 6-9} as ranging from 1.8 to 4. Thus, our results are consistent with a loss of the aging action potential firing phenotype but incompatible with pathological hyperexcitability. Further to this point, we have previously reported¹ that the time to the peak vagal firing rate response to squalamine was about 19 min, and the onset of an increase in vagal firing was 5 min. The increasing vagal firing rate is dependent on prior action on IPANs¹⁰. This is likely to be too rapid for changes in transcription-translation leading to up or down regulation of intracellular proteins¹.

4. The authors proposed a mechanism for squalamine that it can displace the aggregated and misfolded proteins from the cell membrane. Can it be confirmed by imaging those cell membranes for misfolded protein aggregates before and after treatment with squalamine?

Response:

No, we have not confirmed this hypothesis as yet. We do know from studies in *C. elegans* engineered to express a highly aggregating mutant human alpha-synuclein, that the progressive formation of visible aggregates is associated with increasing paralysis; and that administration of squalamine to these animals inhibits aggregation of alpha-synuclein in a dose dependent fashion, resulting in a dose dependent restoration of motility. We have included in the discussion other

recently published biophysical studies that should provide the reader with a sense how squalamine might be acting on the ageing IPAN, with the provision that we have much to do to validate our speculation.

Reviewer #2 (Remarks to the Author):

In this study conducted by Neufeld et al., the researchers investigated the mixed extracellular action potentials of ileum myenteric intrinsic primary afferent neuron (IPAN) and the extracellular discharge from individual single vagal fibres in jejunal mesenteric nerve bundles. A comparison of firing patterns between old and young animals was made. Surprisingly, the authors discovered that infusing jejunal luminal squalamine could restore the vagal firing rates and responses to selective serotonin reuptake inhibitors (SSRIs) in aged mice. This finding is particularly interesting in the context of Parkinson's disease research because Squalamine is known to displace membrane-bound aggregates of misfolded alpha synuclein. The study is well-designed and provides sufficient data.

One minor concern is that the authors should also discuss the broad-spectrum antimicrobial activity of Squalamine in the context of Parkinson's disease-associated constipation.

Response:

An answer to this comment is not straightforward partly because squalamine seems to have direct prokinetic effects on the gut in mouse models of Parkinson's disease because it increases propulsive motility after microbial and faecal contents have been removed from the gut with sterile saline¹¹. Broad-spectrum antibiotics also have complicated and conflicting effects on gut motility. Bacitracin, neomycin, and penicillin V increase mouse colon propulsive motility by direct action on the enteric nervous system^{12, 13} while others such as vancomycin or ampicillin decrease faecal output via action on the microbiome¹⁴.

Certainly there is widespread intestinal dysbiosis in human Parkinson's disease with an overabundance of pro-inflammatory pathogens expressing lipopolysaccharide on their surface¹⁵. Also, the constipation and increase in alpha synuclein observed in alpha synuclein overexpressing mouse constructs is reduced in germfree mice¹⁶. Thus, alterations to the microbiome appear to be associated with Parkinson's disease pathology, although the relationship may be complicated and bidirectional. In general, antibiotics themselves may have anti-inflammatory effects and increase gut motility and have even been suggested to be an alternative to standard Parkinson's disease drugs¹⁷.

In view of the above, it is possible that the broad-spectrum antimicrobial activity of squalamine could contribute to the treatment of Parkinson's disease constipation. However, since squalamine excites enteric primary afferent neurons directly (as shown in the present paper) we lean towards the interpretation that the prokinetic effects of squalamine in Parkinson's disease occurs at least partly by direct action on the enteric nervous system. We have added to the manuscript to cover some of these deliberations.

Reviewer #3 (Remarks to the Author):

The authors describe differences in firing patten and cell membrane characteristics of young- and

old-derived neurons from the enteric and vagal systems.

General comments:

Although the results are somewhat expected (that young- and old-derived neurons show differences), it is important to have a deep characterization of these differences. So I think the article is of interest, however significant ameliorations are needed.

- The n of mice used; and not only fibers analyzed; is of extreme importance, and not cited. Statistical analysis should consider the number of mice used.

Response:

We have added to number of mice used to the text as requested. Parenthetically, results for neurons have always been pooled across animals within each control or treatment group for statistical analysis of these types of experiments in the literature. Representative papers demonstrating such pooling for enteric nervous system or vagus nerve single unit electrophysiology are cited here^{2, 18-22}.

- The use of only males is not in line with "good practices" and guidelines in science (<https://orwh.od.nih.gov/sex-gender/nih-policy-sex-biological-variable>). Therefore I strongly recommend that the authors include females in their study or at least strongly justify why the use of only males. Adding females to the study and reporting on potential differences could be of great interest given the many sex-bias disease related to age and gut-brain axis signaling (e.g. Alzheimer's, Parkinson's...).

Response:

As requested we have added data for females CD-1 mice to this study.

- The text should be revised for mistakes and to improve "fluidity". As it is now this article is hard to read.

Response:

We have edited the text to remove mistakes and spelling errors, and attempted to simplify the text and to break long sentences into shorter ones.

- It would be interesting to see how the young-derived neurons react to the drugs used to treat old-derived neurons.

Response:

This is an important point. We have included new data to show how young IPAN action potential discharge numbers are affected by mucosal application of squalamine or sertraline. Moreover, we have previously described the response of young vagal afferent neuron fibres to squalamine and sertraline², both substances excite the neurons but evoke a unique antidepressant code. Significantly, no response differences were discernable between sexes for these young mice.

- More details about the drugs used in the introduction would be of interest. How does this drug work? Why the authors chose that ones?

Response:

The drugs used/mentioned in the introduction were squalamine and sertraline. Squalamine has pleiotropic activity (multiple therapeutic actions) including antiviral, antibacterial and antifungal activity as well as anti-cancer actions and is therefore likely squalamine acts on cells in a variety of different ways. With regard to our results we think that squalamine, which carries a net positive charge, and has a high affinity for anionic phospholipids reversed the aging code by entering the neurons' cell membrane²³ thus decreasing the negative charge at the intracellular membrane surface. This would depolarise the IPANs making them more excitable and alter their electrophysiological behaviour.

Sertraline is a selective serotonin reuptake inhibitors (SSRI)²⁴.

SSRI increases excitability of enteric sensory neurons via action on the luminal gut epithelium²⁵ which then excites IPAN processes via release of soluble mediators²⁶.

However, we feel that these, as yet to be fully elucidated, transduction mechanisms are beyond the scope of the present paper, which aims to present a unique aging code for afferent signalling with respect to aged vs. young animals.

We chose these drugs because they had previously been used to determine the afferent vagal firing pattern for young animals².

- figure presentation should be improved; colors and consistency (whenever possible make the graph in the same style).

Response:

We have made the colors consistent. Green for young animals, brown for aged, blue for squalamine added, and orange for sertraline added. The same Graphpad Prism style template (bar graphs with individual dot plots and standard errors) have been used for plotting the four parameters describing firing patterns.

The conclusion contains some over-statements, same for the title which is somewhat misleading..

Specific questions/comments:

Response:

We have removed the over-statements and altered the title.

- why only jejunum neurons were analyzed?

Response:

We chose small rather large intestine because the former is more densely innervated by afferent vagal fibres. We have used the jejunum because all the previous recordings from mouse vagal afferent neurons and enteric IPANs have been made using a jejunal segments^{2, 10, 11, 27, 28}. Also, we have found that jejunal rather than duodenum or ileum segments are easier to remove from the attached mesentery and connective tissue with minimal handling from a freshly sacrificed mouse

- Fig 1 not clear from which area are the neurons analyzed.

Response:

We have added “Jejunal” to the legend for Figure 1 to indicate the area analyzed.

- Fig 2 and Sup 1a why this difference in young and old analyzed? It seems like a big difference in group sizes that may result in statistical errors. for Fig1, a dot-plot as done in other figures should be shown here.

Response:

We have modified all figures to include dot plots where appropriate.

About group sizes, we think the reviewer is referring to Fig. 4c where sample size differed by a factor of x3. Sample size differences were smaller for the other parts of Fig 4 and for Fig. 2. Equal sample sizes is not one of the assumptions made in a t test. The t test can handle unequal sample sizes because it takes account of the standard error of the means for each group. So the standard deviation of the group's distribution is divided by the square root of the group's sample size.

Inequality in sample sizes does reduce the power of the t test. In our case power was 0.915 for sample sizes 33 & 77 but would be 0.953 had the sample sizes been equal at 55 for each group. This small reduction in power did not prevent the detection of a statistical difference between the groups.

-Fig 4g and Sup 1d; why there is this "cut" in the graph bar if there is nothing on the top part of the graph?

Response:

We have removed the “cut” as suggested by the reviewer.

-Fig 5c; not clear/described what are the number un the bottom of the columns of the graphs. The number for the dot is useless and only pollute the graph.

Response:

The number at the bottom of all bar graphs (including Fig. 5c) gives the mean value of the parameter being graphed. We do this to aid estimation of the effect size for treatments when viewing the graphs. We have now identified this in the text of figures. The numbers between brackets give the respective sample sizes and we feel this makes the information immediately accessible to the reader.

Comments on the text:

Line 32 - the word reverted is a too strong of statement as some aspects of the aged neurons are not reverted with the drug.

Response:

We have omitted “reverted” as requested.

Line 37 - examples of how these results may have implications in treatments should be added.

Response:

We have added examples as requested.

Lines 78 to 83 - Sentences hard to read and not clear; should be either removed or improved and examples should be added, e.g how could we develop a model of aged ENS?

Response:

We have altered the wording to make these sentences easier to read. Of course, the best model for an aged ENS is aging process itself and the animal model chosen was the mouse.

Lines 101, 103 - ref missing

Response:

We have added the reference, thank you.

Lines 484 to 488 - "over conclusion" remove - no evidence is provided to support the idea of anti-aging effects that can determine the state of the gut microbiota.

Response:

We have removed this paragraph.

References

1. West, C.L., *et al.* Colonic Motility and Jejunal Vagal Afferent Firing Rates Are Decreased in Aged Adult Male Mice and Can Be Restored by an Aminosterol. *Front Neurosci* **13**, 955 (2019).
2. West, C.L., *et al.* Identification of SSRI-evoked antidepressant sensory signals by decoding vagus nerve activity. *Sci Rep* **11**, 21130 (2021).
3. Palmer, J.M., Wong-Riley, M. & Sharkey, K.A. Functional alterations in jejunal myenteric neurons during inflammation in nematode-infected guinea pigs. *Am J Physiol* **275**, G922-935 (1998).
4. Vanden Berghe, P., Kenyon, J.L. & Smith, T.K. Mitochondrial Ca²⁺ uptake regulates the excitability of myenteric neurons. *J Neurosci* **22**, 6962-6971 (2002).
5. Nurgali, K., *et al.* Phenotypic changes of morphologically identified guinea-pig myenteric neurons following intestinal inflammation. *J Physiol* **583**, 593-609 (2007).
6. Nurgali, K., Nguyen, T.V., Thacker, M., Pontell, L. & Furness, J.B. Slow synaptic transmission in myenteric AH neurons from the inflamed guinea pig ileum. *Am J Physiol Gastrointest Liver Physiol* **297**, G582-593 (2009).
7. Foong, J.P., Nguyen, T.V., Furness, J.B., Bornstein, J.C. & Young, H.M. Myenteric neurons of the mouse small intestine undergo significant electrophysiological and morphological changes during postnatal development. *J Physiol* **590**, 2375-2390 (2012).
8. Nurgali, K., Stebbing, M.J. & Furness, J.B. Correlation of electrophysiological and morphological characteristics of enteric neurons in the mouse colon. *J Comp Neurol* **468**, 112-124 (2004).
9. Clerc, N., Furness, J.B., Kunze, W.A., Thomas, E.A. & Bertrand, P.P. Long-term effects of synaptic activation at low frequency on excitability of myenteric AH neurons. *Neuroscience* **90**, 279-289 (1999).
10. Perez-Burgos, A., Mao, Y.K., Bienenstock, J. & Kunze, W.A. The gut-brain axis rewired: adding a functional vagal nicotinic "sensory synapse". *The FASEB Journal* **28**, 3064-3074 (2014).
11. West, C.L., *et al.* Squalamine Restores the Function of the Enteric Nervous System in

- Mouse Models of Parkinson's Disease. *J Parkinsons Dis* **10**, 1477-1491 (2020).
12. Delungahawatta, T., *et al.* Antibiotic Driven Changes in Gut Motility Suggest Direct Modulation of Enteric Nervous System. *Front Neurosci* **11**, 588 (2017).
 13. Forsythe, P., Kunze, W. & Bienenstock, J. Moody microbes or fecal phrenology: what do we know about the microbiota-gut-brain axis. in *BMC Med* **58** (2016).
 14. Ge, X., *et al.* Antibiotics-induced depletion of mice microbiota induces changes in host serotonin biosynthesis and intestinal motility. *J Transl Med* **15**, 13 (2017).
 15. Wallen, Z.D., *et al.* Metagenomics of Parkinson's disease implicates the gut microbiome in multiple disease mechanisms. *Nat Commun* **13**, 6958 (2022).
 16. Sampson, T.R., *et al.* Gut Microbiota Regulate Motor Deficits and Neuroinflammation in a Model of Parkinson's Disease. *Cell* **167**, 1469-1480.e1412 (2016).
 17. Sheng, S., Zhao, S. & Zhang, F. Insights into the roles of bacterial infection and antibiotics in Parkinson's disease. *Front Cell Infect Microbiol* **12**, 939085 (2022).
 18. Krauter, E.M., *et al.* Changes in colonic motility and the electrophysiological properties of myenteric neurons persist following recovery from trinitrobenzene sulfonic acid colitis in the guinea pig. *Neurogastroenterol Motil* **19**, 990-1000 (2007).
 19. Mueller, M.H., *et al.* Extrinsic afferent nerve sensitivity and enteric neurotransmission in murine jejunum in vitro. *Am J Physiol Gastrointest Liver Physiol* **297**, G655-662 (2009).
 20. Rong, W., *et al.* Jejunal afferent nerve sensitivity in wild-type and TRPV1 knockout mice. *J Physiol* **560**, 867-881 (2004).
 21. Mazzuoli, G. & Schemann, M. Mechanosensitive enteric neurons in the myenteric plexus of the mouse intestine. *PLoS One* **7**, e39887 (2012).
 22. Needham, K., *et al.* Identification of subunits of voltage-gated calcium channels and actions of pregabalin on intrinsic primary afferent neurons in the guinea-pig ileum. *Neurogastroenterol Motil* **22**, e301-308 (2010).
 23. Zasloff, M., *et al.* Squalamine as a broad-spectrum systemic antiviral agent with therapeutic potential. *Proc Natl Acad Sci U S A* **108**, 15978-15983 (2011).
 24. Hanson, N.D., Owens, M.J. & Nemeroff, C.B. Depression, antidepressants, and neurogenesis: a critical reappraisal. *Neuropsychopharmacology* **36**, 2589-2602 (2011).
 25. McVey Neufeld, K.A., *et al.* Oral selective serotonin reuptake inhibitors activate vagus nerve dependent gut-brain signalling. *Sci Rep* **9**, 14290 (2019).
 26. Bertrand, P.P. The cornucopia of intestinal chemosensory transduction. *Front Neurosci* **3**, 48 (2009).
 27. West, C.L., *et al.* Microvesicles from *Lactobacillus reuteri* (DSM-17938) completely reproduce modulation of gut motility by bacteria in mice. *PLoS One* **15**, e0225481 (2020).
 28. McVey Neufeld, K.A., Mao, Y.K., Bienenstock, J., Foster, J.A. & Kunze, W.A. The microbiome is essential for normal gut intrinsic primary afferent neuron excitability in the mouse. *Neurogastroenterol Motil* **25**, 183-e188 (2013).

REVIEWERS' COMMENTS:

Reviewer #1 (Remarks to the Author):

My comments are addressed and the manuscript is suitable for publication in Nature Communications Biology. Congratulations to the authors for this effort and valuable piece of science in the gut-brain axis. The gut-brain axis is an emerging field and this manuscript can be of great interest to the scientists in this field to advance aging and medicine research.

Reviewer #3 (Remarks to the Author):

I believe the authors have addressed the major concerns and have improved the article accordingly. The method session have to be updated to include the new mice (females) they added into the study.